# NorMatch: Matching Normalizing Flows with Discriminative Classifiers for Semi-Supervised Learning

**Zhongying Deng**                                                    *zd294@cam.ac.uk*
*Department of Applied Mathematics and Theoretical Physics*
*University of Cambridge*

**Rihuan Ke**                                                    *rihuan.ke@bristol.ac.uk*
*School of Mathematics Research*
*University of Bristol*

**Carola-Bibiane Schönlieb**                                                    *cbs31@cam.ac.uk*
*Department of Applied Mathematics and Theoretical Physics*
*University of Cambridge*

**Angelica I Aviles-Rivero**                                                    *ai323@cam.ac.uk*
*Department of Applied Mathematics and Theoretical Physics*
*University of Cambridge*

**Reviewed on OpenReview:** *https://openreview.net/forum?id=ebiAFpQOLw&noteId=5PmBQKApbT*

## Abstract

Semi-Supervised Learning (SSL) aims to learn a model using a tiny labeled set and massive amounts of unlabeled data. To better exploit the unlabeled data the latest SSL methods use pseudo-labels predicted from *a single discriminative classifier*. However, the generated pseudo-labels are inevitably linked to inherent confirmation bias and noise which greatly affects the model performance. In this work we introduce a new framework for SSL named NorMatch. Firstly, we introduce a new uncertainty estimation scheme based on normalizing flows, as an auxiliary classifier, to enforce highly certain pseudo-labels yielding a boost of the discriminative classifiers. Secondly, we introduce a threshold-free sample weighting strategy to exploit better both high and low confidence pseudo-labels. Furthermore, we utilize normalizing flows to model, in an unsupervised fashion, the distribution of unlabeled data. This modelling assumption can further improve the performance of generative classifiers via unlabeled data, and thus, implicitly contributing to training a better discriminative classifier. We demonstrate, through numerical and visual results, that NorMatch achieves state-of-the-art performance on several datasets.

## 1 Introduction

Deep convolutional neural networks (CNNs) have achieved enormous success in various computer vision tasks (Krizhevsky et al., 2012; Simonyan & Zisserman, 2014; Szegedy et al., 2015; He et al., 2016; Long et al., 2015; Chen et al., 2017; Girshick et al., 2014; Girshick, 2015). The key for such outstanding performance is the large amount of labeled data used in supervised techniques. However, collecting a vast amount of labeled data is time-consuming and labor-extensive. Semi-supervised Learning (SSL) has been a focus of great interest as it mitigates these drawbacks (Tarvainen & Valpola, 2017; Berthelot et al., 2019b;a; Sohn et al., 2020; Li et al., 2021). SSL works under the assumption of learning with a tiny label set and a vast amount of unlabeled data, in which the majority of real-world problems unlabeled data is abundant.

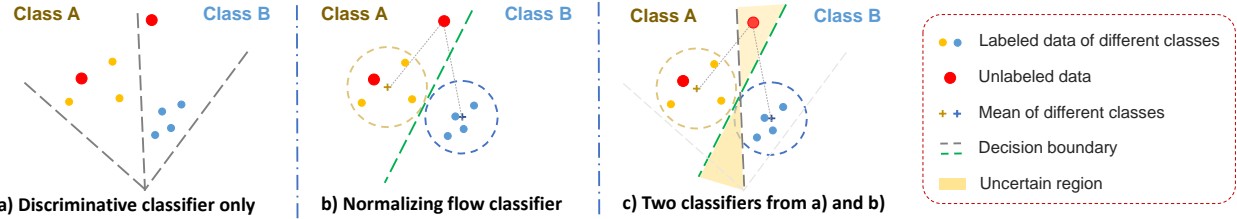

Figure 1: The comparison of a) discriminative classifier, b) normalizing flow classifier (NFC) (Izmailov et al., 2020), and c) discriminative + normalizing flow classifiers in predicting unlabeled data. Here, the data points in all the sub-plots are of the same set of inputs. Our goal is to predict highly certain pseudo-labels for the unlabeled samples (the red dots). Inconsistent predictions from the discriminative classifier and NFC on a sample (e.g., the left red dot) indicate that the pseudo-label is less trustworthy (e.g., the uncertain region in c)). In this case, we will downplay its importance to avoid over-confidence. In contrast, if consistent predictions (e.g., on the right red dot) are achieved among both classifiers, the pseudo labels have higher certainty. Ideally, if the predictions are consistent under any hypothesises, i.e., any different classifiers, we can fully trust the predicted pseudo-labels.

The crucial principle behind SSL is how to better handle the unlabeled set. The current SSL techniques (Sohn et al., 2020; Li et al., 2021; Zheng et al., 2022) use predicted classes from a discriminative classifier as pseudo-labels, for the unlabeled data, with a threshold to filter out low-confidence predictions. The threshold is essentially used to estimate the uncertainty or confidence of the generated pseudo-labels. However, existing threshold-based uncertainty estimation strategies have some disadvantages. Firstly, a manually-set threshold cannot effectively identify noisy samples on which the discriminative classifier can be over-confident. This issue can further cause noise accumulation and the inherent confirmation bias (Tarvainen & Valpola, 2017; Arazo et al., 2020) in pseudo-labeling. Secondly, the threshold is a sensitive hyper-parameter to the performance, which means that a sub-optimal threshold may lead to substantial degradation on some datasets. The intuition is that a high threshold allows for a few pseudo-labels for training while a low threshold introduces high label noise. The optimal threshold depends on the average predicted probability of the discriminative classifier, where the average probability is dependent on the datasets' statistics including the class number and image quality. Finally, thresholding usually discards some low-confidence samples, but they can be hard samples which contribute to better performance.

In this work, we go around the drawbacks associated with thresholding by using normalizing flows to estimate the uncertainty of pseudo-labels from a discriminative classifier. In particular, a Normalizing Flow Classifier (NFC) is used as an auxiliary classifier to estimate the uncertainty of pseudo-labels. The uncertainty estimation is achieved by **match**ing the predictions of a **Nor**malizing flow classifier and the discriminative classifier, which we call NorMatch.

NorMatch uses the NFC to prevent a discriminative classifier from being over-confident on noisy pseudo-labels; as illustrated in Figure 1. This effect is because a pseudo-label having a consensus among diverse classifiers is usually of high quality. Diversity is achieved by using two fundamentally different but complementary classifiers – the NFC as a generative classifier and the Softmax classifier as a discriminative one. NorMatch accepts a pseudo-label if the predicted pseudo-labels of these two classifiers are consistent. Otherwise, NorMatch downplays the importance of such predicted pseudo-label by using the minimum predicted probability of these two classifiers. We call this design Normalizing flow for Consensus-based Uncertainty Estimation (NCUE). NCUE is a threshold-free scheme for different datasets. Moreover, our NCUE in Nor-Match leverages low-confidence samples for model training, which can improve the performance. Overall, our NCUE scheme can effectively tackle the aforementioned three disadvantages of threshold-based uncertainty estimation.

Furthermore, NorMatch also utilizes normalizing flow to model, in an unsupervised fashion, the distribution of unlabeled data. This design is named Normalizing flow for Unsupervised Modeling (NUM). NUM can contribute to learning a better generative classifier on the unlabeled data, thus further improving the performance of a discriminative classifier implicitly. Our contributions are summarized as follows.

- We propose a new SSL method named NorMatch, which utilizes normalizing flows as an auxiliary generative classifier to estimate pseudo-label uncertainty for the discriminative classifier.

- We introduce a threshold-free sample weighting scheme to exploit both high- and low-confidence pseudo-labels called NCUE. We further leverage normalizing flows to model the distribution of unlabeled data in an unsupervised manner (NUM).

- We demonstrate that our NorMatch achieves better, or comparable, performance than state-of-the-art methods on several popular SSL datasets including CIFAR-10, CIFAR-100, STL-10 and Mini-ImageNet.

## 2 Related Work

In this section, we first review the semi-supervised learning methods, then introduce normalizing flow.

### 2.1 Semi-Supervised Learning

Semi-Supervised Learning (SSL) methods can be broadly divided into two categories. The first category adopts consistency regularization while the second one builds upon pseudo-labeling. The idea behind **Consistency regularization** is to enforce consistent outputs, for the same unlabeled sample, under different label-preserving perturbations. These perturbations can be RandAugment (Cubuk et al., 2020), Dropout (Srivastava et al., 2014) or adversarial transformations (Miyato et al., 2018). With multiple perturbed versions of the same sample, Π-Model (Laine & Aila, 2016) minimizes the squared difference between their predictions for a consistent output. Mean Teacher (Tarvainen & Valpola, 2017) further enforces such consistency between the predictions of a model and its exponential moving averaged teacher model. FlowGMM (Izmailov et al., 2020) adopts a normalizing flow model together with a Gaussian Mixture Model to enforce a probabilistic consistency regularization.

Unlike FlowGMM which uses a single normalizing flow to encode the clustering principle (no discriminative classifier included), our NorMatch uses it as an auxiliary classifier to deal with the threshold-based uncertainty estimation problem caused by a single discriminative classifier. We estimate the uncertainty of pseudo-labels based on the consensus of these two classifiers. Based on the uncertainty, we propose a threshold-free sample weighting scheme to assign different weights for pseudo-labels. As a result, NorMatch significantly outperforms FlowGMM (see Table 6).

**Pseudo-labeling**, including self-training, uses the model's predictions as pseudo-labels for the unlabeled data, with the pseudo-labels used for the model training in a supervised fashion. MixMatch (Berthelot et al., 2019b) generates 'soft' pseudo-labels using the averaged prediction of the same image with multiple strong augmentations while ReMixMatch (Berthelot et al., 2019a) uses weakly-augmented ones to obtain pseudo-labels. It further proposes a distribution alignment to encourage the distribution of pseudo-labels to match that of ground-truth labels of labeled data. FixMatch (Sohn et al., 2020) also employs weak augmentation for pseudo-label generation but it obtains the one-hot 'hard' pseudo-labels. Since 'hard' pseudo-labels may contain noise, it further introduces a threshold to filter out low-confidence thus potentially noisy samples. To improve the pseudo-labels strategy of FixMatch, CoMatch (Li et al., 2021) further imposes a smoothness constraint on the pseudo-labels by introducing an extra contrastive learning task. SemCo (Nassar et al., 2021) improves the pseudo-labels by adopting two discriminative classifiers for co-training. Some other methods seek to improve the pseudo-label by modifying the threshold. For example, Dash (Xu et al., 2021) improves FixMatch by proposing an adaptive threshold which decreases during training. Adsh (Guo & Li, 2022) argues that a fixed threshold for all the classes is sub-optimal, so it designs adaptive thresholds for different classes to improve over FixMatch. A few works have explored how to improve pseudo-labelling through the lens of graphs, where different types of Laplacian energies have been used. The CREPE model (Aviles-Rivero et al., 2019) introduced a new energy model based on the graph 1-Laplacian, which generates highly certain pseudo-labels. LaplaceNet (Sellars et al., 2022) uses quadratic energy along with a new multi-sample augmentation scheme.

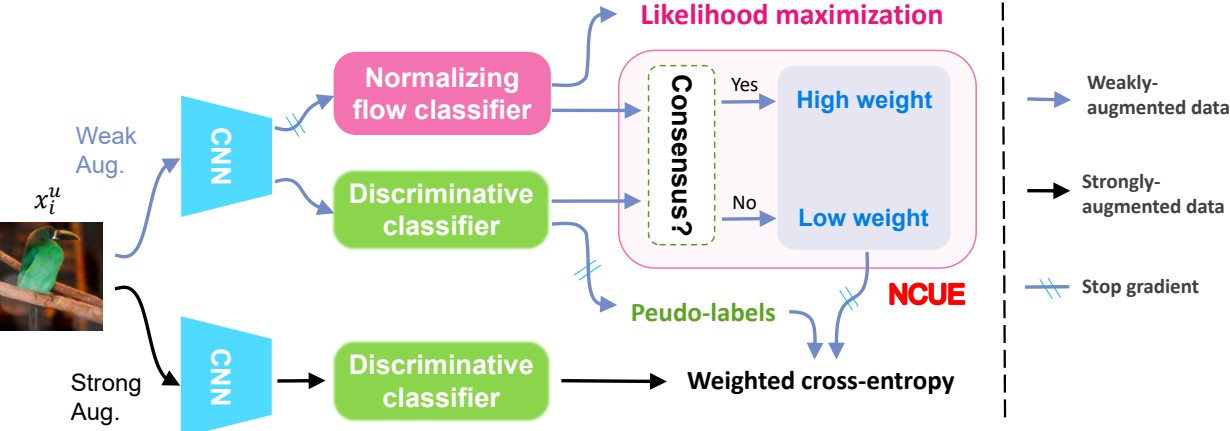

Figure 2: The overview of our NorMatch for unlabeled data. The modules with the same colors (i.e., the CNN and discriminative classifier) share the same set of parameters. NorMatch uses the shared CNN backbone to extract the features of weakly- and strongly-augmented versions of the same unlabeled sample. Then the weakly-augmented features are input to the Normalizing Flow Classifier (NFC) for Unsupervised Modeling (NUM) by likelihood maximization, and to the discriminative classifier to obtain the pseudo-labels. These features are also input to the NFC and the discriminative classifier to enforce a Consensus Uncertainty Estimation (called NCUE). The NCUE generates weights for each sample/pseudo-label, which highlights the consistent predictions and downplays the disagreed ones. The weights together with pseudo-labels are then used to enforce a weighted cross-entropy, which supervises the training for the strongly-augmented version.

Our NorMatch also builds on FixMatch, but leverages the consensus of an auxiliary generative classifier and the main discriminative one to improve pseudo-labels. Importantly, NorMatch is threshold-free, and simpler yet more effective than existing methods.

## 2.2 Normalizing Flow

Normalizing Flows (Dinh et al., 2014; 2016; Kobyzev et al., 2020) composes some invertible and differentiable mapping functions to transform a simple distribution, e.g., standard Gaussian, to match a complex one, e.g., the distribution of real data. Such mappings preserve the exact likelihood, which facilitates the probability density estimation for new data. This property of normalizing flow can be used as a generative classifier, which cannot be achieved by other generative models, such as generative adversarial network (GAN) (Goodfellow et al., 2020) and variational auto-encoder (VAE) (Kingma & Welling, 2013). As a generative model, normalizing flow can model the marginal distribution of the real data by likelihood maximization. This makes it suitable for unsupervised tasks since no ground-truth labels are needed for model training.

Generative models are widely used to generate data or enforce consistency for SSL. However, scarce works use generative models, especially normalizing flow, as a classifier to help the discriminative one for uncertainty estimation which is the major challenge in pseudo-label-based SSL. To this end, we exploit the normalizing flow classifier (NFC) for uncertainty estimation. Remarkably, our NorMatch exploits NFC (Izmailov et al., 2020) to weigh each pseudo-label and uses it to model the marginal distribution of unlabeled data, both contributing to better performance for SSL task.

## 3 Methodology

In this section, we detail our motivation on leveraging the Normalizing Flow Classifier (NFC) as an auxiliary classifier to estimate the uncertainty of pseudo-labels predicted from the discriminative classifier. We then present how the NFC-based NorMatch works for unlabeled data. Finally, we illustrate the training and inference process of our proposed method.

### 3.1 Motivation

Our insight is that the consensus among diverse classifiers on a pseudo-label can reduce the risk of confirmation bias. Following the work of (Melville & Mooney, 2003), we can define the diversity as the measure of disagreement across different classifiers. Diversity is ensured by using two fundamentally different but complementary classifiers, namely, the NFC as a generative classifier and the Softmax classifier, e.g., a fully connected layer followed by a Softmax activation function, as a discriminative one.

We choose the Normalizing Flow Classifier (NFC) as the auxiliary classifier for the following reasons. (1) Compared to other generative models like GAN or VAE, normalizing flows can evaluate the exact probability density for new test data. This means that given new test data, we can know the probability of the data following the distribution of a specific class. In practice, the class-specific distribution is modeled by the $y$-th component of the Gaussian Mixture Model (GMM) in (1) (as we implement the NFC as a RealNVP (Papamakarios et al., 2017) followed by a GMM prior (Izmailov et al., 2020), with the RealNVP acting as invertible and differentiable mapping functions). If we further normalize such a class-specific probability using all the probabilities of all the classes, then the output of the Normalizing Flow can be used to measure the probability of the input data following the distribution of all the different classes, which is illustrated in (1). Therefore, normalizing flow can be used as a generative classifier while the others cannot. (2) Compared to another Softmax-based discriminative classifier as an auxiliary one, the NFC can be used to model the marginal distribution of unlabeled data in an unsupervised way. This can boost the performance (as demonstrated in Table 2). More importantly, NFC, as a generative classifier, is more complementary to the main discriminative classifier than using another discriminative classifier. We then can ensure the diversity of these two classifiers.

Concretely, diversity manifests in the following ways. Firstly, the Normalizing Flow Classifier (NFC) predicts the conditional probability derived from Bayes Theorem while a discriminative classifier directly learns the conditional probability distribution $p(y|z)$, where $z$ is the feature representation of an image $x$ and $y$ is an element of the label space. Secondly, NFC is an Euclidean distance-based classifier (Izmailov et al., 2020) while the Softmax-based discriminative classifier focuses on cosine distance. The NFC is Euclidean distance-based classifier as it predicts the labels based on the following conditional probability:

$$p_n(y|z) = \frac{\mathcal{N}(z|\mu_y, \Sigma_y)}{\sum_{k=1}^{C} \mathcal{N}(z|\mu_k, \Sigma_k)} \propto \mathbf{E}(||z - \mu_y||_2^2), \tag{1}$$

where the denominator $\sum_{k=1}^{C} \mathcal{N}(z|\mu_k, \Sigma_k)$ is a normalization factor shared by all the class. $C$ is the class number, and $\mathcal{N}(\mu_y, \Sigma_y)$ denotes the $y$-th class/component in a Gaussian Mixture Model (GMM), parameterized by the mean $\mu_y$ and covariance $\Sigma_y$. It shows that the probability is influenced by the Euclidean distance between a sample's feature $z$ and the mean $||z - \mu_y||_2^2$. It is worth noting that to facilitate a better understanding, (1) simplifies the NFC by viewing its invertible and differentiable mapping functions $f_n(\cdot)$ as an identity matrix $I$, i.e., $f_n(\cdot) = I$. This simplification makes the Normalizing Flow degrade to a Gaussian Mixture Model (GMM), which is easier to understand. But in practice, we use a RealNVP (Papamakarios et al., 2017) as the invertible and differentiable functions $f_n(\cdot)$, so the input of (1), i.e., $z$, is actually transformed to $f_n(z)$ before input to GMM. This further leads to $p_n(y|z) \propto \mathbf{E}(||f_n(z) - \mu_y||_2^2)$.

In contrast to a Normalizing Flow Classifier (NFC), a discriminative classifier makes predictions based on conditional probability, which reads:

$$p_d(y|z) = \frac{\exp(W_y^T z)}{\sum_{k=1}^{C} \exp(W_k^T z)} \propto W_y^T z = ||W_y^T|| \cdot ||z|| \cdot \cos(W_y^T, z), \tag{2}$$

where $W_y$ is the weight for class $y$. It is clear that $p_d(y|z)$ is based on the cosine similarity of $W_y^T$ and $z$. Intuitively, two cosine distance-based discriminative classifiers are less diverse than Euclidean distance-based NFC combined with a cosine distance-based discriminative classifier. Furthermore, the Euclidean distance of NFC is calculated between the $f_n(z)$ and the mean feature of the $y$-th class, $\mu_y$. In contrast, the Softmax classifier computes the cosine similarity between the **latent feature** $z$ and **the weight of the $y$-th component of the classifier** $W_y$ as in (2), i.e., $z$ is directly used to compute the labels rather than

transformed by any invertible and differentiable functions. Since $f_n(z)$ in the NFC is not equal to $z$ in the Softmax classifier and the mean feature of the $y$-th class (i.e., $\mu_y$) is different from the weight of the $y$-th component of the classifier (i.e., $W_y$), these two classifiers are considered to be sufficiently diverse.

Lastly, diversity comes from the lens of statistical learning where a generative model has a higher asymptotic error than the discriminative one. However, the generative one can reach the asymptotic error much faster. That is, our model enforces these two distinctive performance regimes as complementary– this is translated to enforce higher diversity in terms of boundary between classes (discriminative) while also the distribution of individual classes.

As illustrated in Figure 1, if consistent predictions are achieved among these diverse classifiers, i.e., under different measurements (Euclidean and cosine), the predicted pseudo-labels have higher certainty. Otherwise, the pseudo-label is less reliable, thus its importance should be downplayed. We remark that we downplay low-confidence pseudo-labels rather than simply ignore them as current methods do (Sohn et al., 2020; Li et al., 2021). We do this because they can be hard samples, which might contribute to better performance.

## 3.2 NorMatch

The key in our NorMatch is to exploit the discriminative classifier and Normalizing flows for Consensus-based Uncertainty Estimation (NCUE), and apply the Normalizing flow for Unsupervised Modeling (NUM), as depicted in Figure 2. NCUE estimates the uncertainty for pseudo-labels by emphasizing consistently predicted pseudo-labels and downplaying low-confidence ones that cause disagreement. NUM uses the Normalizing Flow Classifier (NFC) to model the distribution of unlabeled data by likelihood maximization. We detail these two designs next.

### 3.2.1 Normalizing flow for Consensus-based Uncertainty Estimation (NCUE)

As shown in Figure 2, the unlabeled data $\mathcal{D}_u = \{x_i^u\}_{i=0}^{N_u}$ where $N_u$ is the total sample number, are applied with both weak and strong augmentations to obtain two different versions of the same input. For weakly-augmented versions, we use flipping and cropping (still denote it as $x_i^u$). For the strongly-augmented version, we have $\mathcal{A}(x_i^u)$, being $\mathcal{A}$ RandAugment (Cubuk et al., 2020). These two versions are then input to a CNN backbone to extract features for classification. The feature of the weakly-augmented version is fed to the Normalizing Flow Classifier (NFC) and the discriminative one to obtain the probabilities $p_n(y|x_i^u), p_d(y|x_i^u)$, as in (1) and (2) respectively (denoted as $p_n, p_d$ for clarity). We can then obtain the pseudo-label from the discriminative classifier as $\hat{y}_i^u = \arg\max(p_d)$ or $\hat{y}_i^u = p_d$. The latter is not a one-hot version as the latest methods use it to enforce distribution alignment (Berthelot et al., 2019a; Li et al., 2021). The design choice is evaluated in the experiments.

With these probabilities $p_n, p_d$, NCUE estimates the uncertainty of a pseudo-label by investigating the consensus of the Normalizing Flow Classifier (NFC) and the discriminative classifier, and then adaptively sets a weight for such pseudo-label. Formally, the NCUE reads:

$$\tau(x_i^u) = \begin{cases} 1, & \text{if } \arg\max(p_d) = \arg\max(p_n), \\ \min(p_d, p_n), & \text{if } \arg\max(p_d) \neq \arg\max(p_n), \end{cases} \tag{3}$$

being $\tau(x_i^u)$ the weight for each unlabeled sample. It means that we accept the pseudo-label if it achieves consensus among these two classifiers. Otherwise, we downplay its importance by $\min(p_d, p_n)$ as it can be noise. With $\tau(x_i^u)$ as sample weight, and $\hat{y}_i^u$ as the pseudo-label, the loss for the unlabeled data, i.e., the weighted cross-entropy in Figure 2, is given by:

$$L_u(\theta_d) = \frac{1}{\mu B} \sum_i^{\mu B} \tau(x_i^u) \cdot H(\hat{y}_i^u, p_d(y|\mathcal{A}(x_i^u), \theta_d)), \tag{4}$$

where $p_d(y|\mathcal{A}(x_i^u), \theta_d)$ is the probability of strongly-augmented version $\mathcal{A}(x_i^u)$ predicted from the discriminative classifier. $\theta_d$ is the parameters of the CNN backbone and the discriminative classifier. $B$ is batch size, $\mu = 7$ as in (Sohn et al., 2020), and $H(y, p)$ is the cross-entropy.

### 3.2.2 Normalizing flow for Unsupervised Modeling (NUM)

NUM models the distribution of the features $z$ of unlabeled data $x$ by likelihood maximization estimation. The likelihood for the feature of $i$-th unlabeled image is

$$p_n(z_i^u) = \sum_c^C p_n(z_i^u|y = c)p(y = c), \tag{5}$$

where $p_n(z_i^u|y = c)$ is obtained by feeding the feature $z_i^u$ of $x_i^u$ to the $c$-th class/component of the GMM. We then can optimize the parameters of normalizing flow $\theta_n$ to maximize the joint probability of unlabeled data

$$p_n(\mathcal{D}_u|\theta_n) = \prod_i^{N_u} p_n(z_i^u|\theta_n). \tag{6}$$

Equivalently, we can achieve the maximization of (6) by minimizing the negative log-likelihood of $p_n(\mathcal{D}_u|\theta_n)$. Therefore, we define a loss function $L_u(\theta_n)$ for the goal of likelihood maximization in Figure 2. $L_u(\theta_n)$ is formulated as:

$$L_u(\theta_n) = -\log p_n(\mathcal{D}_u|\theta_n). \tag{7}$$

**Remark.** We model the probability mass of the latent features $p(z)$, rather than the original input images $p(x)$, using Normalizing Flow. In NUM, we input the latent feature $z_i^u$ to the Normalizing Flow Classifier (NFC) to obtain $p_n(z_i^u|y = c)$. Then the invertible and differentiable mapping functions (implemented as RealNVP (Papamakarios et al., 2017)) in the NFC can be learned to match the complex distribution of $z_i^u$ by optimizing (7). Since we use the invertible and differentiable mapping functions, denoted as $T : \mathbb{R}^n \rightarrow \mathbb{R}^n$, to transform a simple GMM $p(g)$ to match a more complex distribution of $p(z)$, the NFC in our method is Normalizing Flow with the $p(z)$ computed by

$$p(z) = p(g)|\det \frac{\partial T^{-1}(z)}{\partial z}| = p_g(T^{-1}(z))|\det \frac{\partial T^{-1}(z)}{\partial z}| \quad \text{where } g = T^{-1}(z). \tag{8}$$

It is notable that $T = f_n^{-1}$, where $f_n$ is defined in Section 3.1. Furthermore, we remark that the input of Normalizing Flow is not necessarily to be the original images $x$ but can also be the latent features $z$ if we regard the latent features as a complex distribution. Here, the "complex distribution" is a relative concept, which means that the distribution of latent features is usually more complex than GMM. We adopt the latent features $z$ as the input of the Normalizing Flow rather than the images $x$ for two reasons: 1) we aim to use the Normalizing Flow as a generative classifier, which usually takes semantic features as its input for better performance. Thus, using Normalizing Flow to model latent features $z$, containing more semantic information than the images $x$, can better achieve our goal of improving classification accuracy; 2) The images $x$ are in high dimension (e.g., $3 \times 96 \times 96 = 27,648$ dimension on Mini-ImageNet dataset) and Normalizing Flow cannot reduce their dimension (otherwise the loss of dimension/information can make the Normalizing Flow NOT invertible). In this case, directly modeling $p(x)$ in a high-dimensional image space costs too much computational resources which we cannot afford.

### 3.3 Training and Inference Schemes

For the labeled samples $\{x_i^l, y_i\}_{i=0}^{N_l}$, we adopt cross-entropy for supervised training. Formally, the Normalizing Flow Classifier (NFC) $\theta_n$ is trained on a labeled set by minimizing

$$L_x(\theta_n) = \frac{1}{B} \sum_i^B H(y_i, p_n(y|x_i^l, \theta_n)), \tag{9}$$

where $p_n(y|x_i^l, \theta_n)$ is the probability distribution of the sample $x_i^l$. We also define $L_x(\theta_d)$ as the supervised loss for the discriminative classifier.

Our total training loss is then formulated as:

$$L = L_x(\theta_d) + L_u(\theta_d) + L_x(\theta_n) + \lambda L_u(\theta_n), \tag{10}$$

where $\lambda$ is a hyper-parameter. Note that the gradients of $L_u(\theta_n)$ and $L_x(\theta_n)$ are only back-propagated to NFC (i.e., $\theta_n$) rather than the CNN backbone because we discard the auxiliary Normalizing Flow Classifier (NFC) during inference. In this case, these two loss terms contribute to feature learning by influencing pseudo-labels. Concretely, they influence the learning of NFC, which impacts the weight $\tau$ of pseudo-labels. $\tau$ in (4) can adjust the gradient of $L_u(\theta_d)$ to the CNN for feature learning and contribute to better performance. We found that this design worked the best (see Table 3).

For inference, we only use the discriminative classifier while discard the NFC.

## 4 Experiments

We conduct extensive experiments on CIFAR-10, CIFAR-100, STL-10 and Mini-ImageNet to demonstrate the effectiveness of our NorMatch.

### 4.1 Experimental Setting

**Dataset and Protocols.** (1) *CIFAR-10* (Krizhevsky et al., 2009) has 10 classes, each with 5,000 images of size 32×32 for training, and 1,000 images for testing, so there are 60,000 images in total. Following (Sohn et al., 2020), we evaluate our methods in the settings of training with 4, 25, and 400 labels per class, respectively. (2) *CIFAR-100* (Krizhevsky et al., 2009) has the same image size as CIFAR-10, but comprises 100 classes. Each class includes 500 images for training and 100 for testing. We also follow (Sohn et al., 2020) to report the results of our models trained on 4, 25, and 100 labels per class, respectively. (3) *STL-10* (Coates et al., 2011) consists of 96×96 images of 10 classes, with 500 training and 800 test images per class. We train our model on 1000 labels, with 100 for each class, following (Sohn et al., 2020). (4) *Mini-ImageNet* is a subset of ImageNet (Russakovsky et al., 2015), which includes 84×84 images from 100 classes, with 600 images per class. We adopt the training and testing split from (Iscen et al., 2019), then evaluate NorMatch in the settings of 40 labels for each class.

**Implementation Details.** The Softmax classifier is implemented as a fully connected layer followed by a Softmax activation function. The backbone CNN (Cf. the blue block in Figure 2) is selected as follows. We follow (Sohn et al., 2020) to adopt Wide ResNet-28-2 for CIFAR-10 and Wide ResNet-28-8 (Zagoruyko & Komodakis, 2016) for CIFAR-100. On STL-10 and Mini-ImageNet, we use a ResNet-18 (He et al., 2016) as the backbone CNN, following (Li et al., 2021) and (Nassar et al., 2021), respectively. The other training settings for all these datasets are the same (unless otherwise specified). Specifically, we optimize the model using Stochastic Gradient Descend (SGD) with Nesterov momentum (Sutskever et al., 2013). The initial learning rate is 0.03 and then decreases according to a cosine learning decay (Loshchilov & Hutter, 2016). The batch size $B$ is 64 and the total training iteration is $2^{20}$ (1024 epochs with each epoch having 1024 iterations, except on Mini-ImageNet training for 600 epochs). We follow the latest works (Berthelot et al., 2019a; Li et al., 2021) to use distribution alignment to $\hat{y}_i^u$ in (4). We do not apply sharpening or one-hot to $\hat{y}_i^u$ (except on STL-10 where a one-hot version is used). We use the exponential moving average of model parameters to report the final performance, as most SSL methods (Berthelot et al., 2019a; Sohn et al., 2020; Li et al., 2021) do.

For the settings specific to our NorMatch, we set the default value of $\lambda$ in (10) to 1e-6 for all the datasets. The NFC is a RealNVP (Papamakarios et al., 2017) (with 6 coupling layers) followed by a Gaussian Mixture Model (GMM) prior (Izmailov et al., 2020). As such, $\theta_n$ in (10) includes three parts: a) weights of GMM, initialized as 1 for each class, b) the $\mu$ (init. as 0) and $\Sigma$ (init. as 1) of GMM, and c) randomly initialized coupling layers of the Normalizing Flow Classifier (NFC). It is trained with AdamW (Loshchilov & Hutter, 2017) optimizer using an initial learning rate of 0.001 with a cosine decay.

Our implementation is based on PyTorch (Paszke et al., 2019) and our code is available at `https://github.com/Zhongying-Deng/NorMatch`.

Table 1: Ablation study on CIFAR-10 with 40 labels. NCUE and NUM are proposed in Section 3.2.1 and 3.2.2, respectively. The NCUE variant sets the weight to 0 when the NFC's predictions are different from the discriminative classifier's, i.e., it simply discards all the low-confidence pseudo-labels. The best result is highlighted in yellow.

| Methods | Accuracy |
|---|---|
| Baseline (FixMatch (Sohn et al., 2020)) | 87.77 |
| Baseline + NCUE | 92.78 |
| Baseline + NCUE variant | 91.55 |
| **NorMatch** (Baseline + NCUE + NUM) | **93.41** |

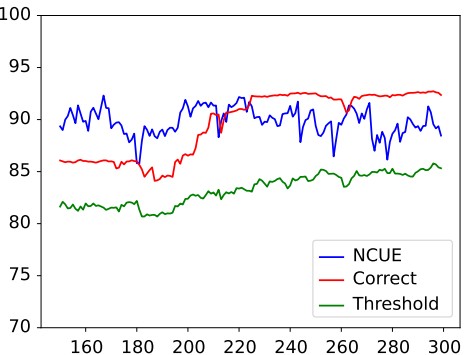

Figure 3: The amount of low-uncertainty (high-confidence) pseudo-labels obtained by NCUE (the blue curve) and threshold-based FixMatch (the green curve with the threshold set to 0.95 as in (Sohn et al., 2020)), respectively. The red curve denotes the number of correct pseudo-labels measured by using ground-truth labels. The x-axis represents the training epoch while the y-axis denotes the percentage (%) of total samples.

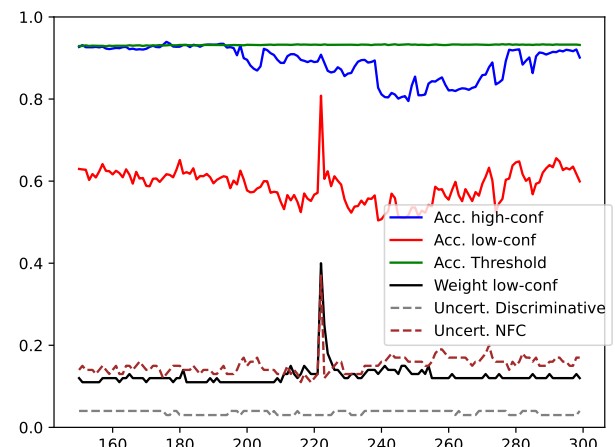

Figure 4: Visualization of 1) the accuracy of high/low-confidence predictions (the blue and red curves) and the weight distributions (the dark curve) of our NorMatch; 2) the accuracy of predictions from the threshold-based FixMatch, i.e., the green curve; 3) the uncertainties of discriminative classifier and NFC (dashed gray and brown curve respectively) of our NorMatch.

## 4.2   Delving into NorMatch Performance

In this section, we present a comprehensive analysis on CIFAR-10 with 40 labels to better understand each module in our method. Our baseline model is the vanilla FixMatch with threshold and one-hot pseudo-labels. For this analysis, we only run 300 epochs to save time.

**Effectiveness of NCUE.** Table 1 shows that the Normalizing flow for Consensus-based Uncertainty Estimation (NCUE, proposed in Section 3.2.1) significantly improves the FixMatch baseline by 5.01%. Note that FixMatch uses a threshold-based uncertainty estimation, so the superiority of NCUE verifies that using the consensus among NFC and the discriminative classifier can be better than using a threshold to estimate uncertainty for pseudo-labels.

Furthermore, to investigate whether we should simply discard all the low-confidence pseudo-labels, we evaluate an NCUE variant which sets the weights of low-confidence samples to 0 (rather than $\min(p_d, p_n)$ as in (3)). This variant (the 3rd row) decreases the performance by 1.23%. The degradation implies that simply ignoring all the low-confidence samples can be sub-optimal, as they can be hard samples and contribute to better performance.

**Further analysis on NCUE.** To better understand how our NCUE works, we further provide the following visualizations. 1) Figure 3 plots the amount of low-uncertainty (or high-confidence) pseudo-labels. The

results are obtained by running the proposed training scheme, and then using the generated pseudo-labels to compare NCUE and thresholding. We find that the amount of high-confidence pseudo-labels from NCUE (the blue curve) is very similar to that of the correct pseudo-labels. Thus, our NCUE can adaptively choose a proper amount of high-confidence pseudo-labels for training. In contrast, a fixed threshold of 0.95 (the green curve) ignores too many samples that have correct pseudo-labels. This comparison explains why our NCUE works better than a fixed threshold, which is widely used in FixMatch-based methods (Nassar et al., 2021; Hu et al., 2021).

2) Figure 4 depicts the accuracy of low-uncertainty (or high-confidence) pseudo-labels obtained by NCUE (the blue line) and a fixed threshold (the green line denoted as 'Acc. Threshold') respectively. We find that the pseudo-labels' accuracy of these two is similar ($\sim$93%), but NCUE has a larger absolute number of correct pseudo-labels as it has a much higher recall rate (i.e., more high-confidence pseudo-labels as in Figure 3, about 92% vs. 85%). That is, for NCUE, 50K training images with 92% high-confidence pseudo-labels, among which $\sim$93% are accurate, totally 50K$\times$92%$\times$93%$\approx$42.8K accurate pseudo-labels. In contrast, a fixed threshold has only 50K$\times$85%$\times$93%$\approx$39.5K accurate pseudo-labels, thus achieving inferior performance.

3) Figure 4 also provides the percentage of correct predictions within high- and low-confidence samples (the blue and red curves respectively), as well as the weight distributions of low-confidence samples (the dark line). These curves can show how the consensus re-weighting technique in NCUE works. We observe that the high-confidence samples have high accuracy, which is essential for good performance; the low-confidence samples have low accuracy and small weights. Small weights can alleviate the issue of over-confidence or confirmation bias, and meanwhile, take full use of low-confidence samples for better performance. The advantage of small weights is also verified in Table 1 where small weight-based NCUE achieves 92.78%, outperforming the NCUE variant by 1.23% which simply ignores these low-confidence samples.

4) Finally, Figure 4 visualizes the uncertainty of the discriminative classifier and NFC (dashed gray and brown lines). We measure the uncertainty of these classifiers by using $1 - p_{all}$, with $p_{all}$ being the average predicted probabilities of a classifier for all the samples. It can be seen that the discriminative classifier in NorMatch can be more over-confident (lower uncertainty) compared to NFC, thus may lead to confirmation bias. NFC is less over-confident (i.e., higher uncertainty) to alleviate confirmation bias.

**Importance of NUM.** In Table 1, we can also see that the Normalizing flow-based Unsupervised Modeling (NUM, proposed in Section 3.2.2) brings a performance gain over FixMatch + NCUE. With NUM, the normalizing flow is exposed to unlabelled data, in comparison to the case without NUM where the NFC is trained using only the labelled data. In particular, the NFC is based on the calculation of the conditional probability $p_n(y|x)$, hence having the unlabeled data for training potentially enforces better prediction of the labels.

**NFC vs. an auxiliary discriminative classifier.** We argue that the Normalizing Flow Classifier (NFC) is more diverse and complementary to the main discriminative classifier. Here, we replace the NFC with a discriminative classifier to justify our argument. For fair comparison, the replacement classifier is with similar parameters (also 6 layers, each layer comprising a fully-connected layer followed by ReLU and batch normalization (Ioffe & Szegedy, 2015)) to NFC. We show their parameters and performance in Table 2. Our NFC is better than using another discriminative classifier by about 1%, demonstrating that NFC is more complementary. Another notable observation is that the NFC is lightweight, with only 0.08M parameters and 0.08M Multiply ACcumulate operations (MACs). This shows its efficiency.

**DC + NFC vs. two NFCs.** To further support the argument that the discriminative classifier (DC) and the Normalizing Flow Classifier (NFC) are the better options for diversity, we also replace the main discriminative classifier with an NFC. This design choice leads to two NFCs. Note that in this case, the gradient of one of these two NFCs needs to back-propagate to the backbone CNN so that the backbone CNN can be updated. From the last row of Table 2, we observe that two NFCs cause a large performance drop of 12.60% when compared with our default setting (DC + NFC). The drop is probably because the diversity of two NFCs is not as large as DC + NFC. While in NorMatch, the diversity of classifiers plays a vital role.

Table 2: Evaluation on different classifier combinations. Normalizing Flow Classifier (NFC) vs. another Discriminative Classifier (DC). MACs: Multiply ACcumulate operations. The #Param and MACs denote the additional parameters and MACs that the extra classifier introduces.

| Methods | #Param | MACs | Accuracy |
|---|---|---|---|
| DC + NFC (Default setting) | 0.08M | 0.08M | **93.41** |
| DC + Another DC | 0.09M | 0.09M | 92.42 |
| NFC + NFC | 0.08M | 0.08M | 80.81 |

Table 3: Evaluation on stopping gradient of the Normalizing Flow Classifier (NFC) to CNN backbone. $r_{hc}$ denotes the high-confidence samples that have their predicted probability $>0.95$.

| Stop gradient | $r_{hc}$ | Accuracy |
|---|---|---|
| ✓ | 82.31 | **93.41** |
| ✗ | 32.82 | 29.27 |

**Necessity of stopping gradient of NFC to the backbone CNN.** As stated in Section 3.3, during training, the gradients of $L_u(\theta_n)$ and $L_x(\theta_n)$ are only back-propagated to NFC ($\theta_n$) rather than the CNN backbone (please also see the stop gradient symbol in Figure 2). We thus evaluate this design choice in Table 3. We can see that if we allow the gradient to be back-propagated to the CNN backbone and hence play a role in its parameter updates, the training almost fails (with a poor accuracy of 29.27%). This is probably because the gradient from the NFC may harm the discriminative feature learning supervised by the main classifier. As a result, the features from the backbone CNN can hardly fit these two fundamentally different classifiers simultaneously. This can be inferred from the decreased amount of high-confidence samples, e.g., from 82.31% to 32.82%. The sharp decrease is because the features are not discriminative enough to achieve high confidence, further causing poor performance.

**Pseudo-Labels or No Pseudo-Labels to train NFC on unlabeled data?** We further investigate whether the performance can be improved by training the Normalizing Flow Classifier (NFC) with pseudo-labels on unlabeled data. Table 4 shows that pseudo-label-based supervised training for NFC decreases the performance (the first two rows). This is probably because the pseudo-labels can contain noise, which makes the NFC less effective. In addition, since both the NFC and the main discriminative classifier are trained with the same set of pseudo-labels, they may suffer from the same set of noise, thus not complementary to each other anymore. As such, noisy pseudo-labels cannot be correctly identified based on the consensus of these two classifiers. This can further lead to confirmation bias.

**More NFCs are Better Performance?** It is natural to ask whether one more Normalizing Flow Classifier (NFC) as an auxiliary classifier can further help. To answer this question, we conduct the experiment by introducing an extra NFC to our NorMatch, leading to two NFCs with the same architecture. The Normalizing flow for Consensus-based Uncertainty Estimation (NCUE) is then enforced on these two NFCs and the discriminative classifier in a similar way to equation 3: If and only if these three classifiers predict the same pseudo-label for an unlabeled sample, the weight of such a sample is 1; Otherwise, the weight is the minimal probability of these three predictions. We then show its result in the last row of Table 4. We

Table 4: Evaluations on (1) using pseudo-labels to train Normalizing Flow Classifier (NFC) on unlabeled data, and (2) one more NFC as the auxiliary classifier.

| Methods | Accuracy |
|---|---|
| Default setting | **93.41** |
| Use pseudo-label to train NFC | 92.72 |
| One more NFC as auxiliary classifier | 92.67 |

Table 6: Classification accuracy (%) on CIFAR-10, CIFAR-100 and STL-10. Best results are in bold.

| | CIFAR-10 | | | CIFAR-100 | | | STL-10 |
|---|---|---|---|---|---|---|---|
| **Methods** | 40 labels | 250 labels | 4000 labels | 400 labels | 2500 labels | 10000 labels | 1000 labels |
| Π-Model | - | 45.74±3.97 | 58.99±0.38 | - | 42.75±0.48 | 62.12±0.11 | - |
| Mean Teacher | - | 67.68±2.30 | 90.81±0.19 | - | 46.09±0.57 | 64.17±0.24 | - |
| MixMatch | 52.46±11.50 | 88.95±0.86 | 93.58±0.10 | 32.39±1.32 | 60.06±0.37 | 71.69±0.33 | 38.02±8.29 |
| ReMixMatch | 80.90±9.64 | 94.56±0.05 | 95.28±0.13 | 55.72±2.06 | 72.57±0.31 | 76.97±0.56 | - |
| FlowGMM-cons | - | - | 80.9 | - | - | - | - |
| FixMatch | 86.19±3.37 | 94.93±0.65 | 95.74±0.05 | 51.15±1.75 | 71.71±0.11 | 77.40±0.12 | 65.38±0.42 |
| CoMatch | 93.09±1.39 | 95.09±0.33 | - | - | - | - | 79.80±0.38 |
| SemCo | - | 94.88±0.27 | **96.20±0.08** | - | 68.07±0.01 | 75.55±0.12 | - |
| Dash | 86.78±3.75 | **95.44±0.13** | 95.92±0.06 | 55.24±0.96 | 72.82±0.21 | 78.03±0.14 | - |
| NorMatch (Ours) | **94.70±0.16** | 95.06±0.18 | 95.89±0.12 | **59.39±0.39** | **73.41±0.29** | **78.55±0.18** | **81.38±0.12** |

observe a performance drop with one more NFC. This means that using a single NFC can already work well for uncertainty estimation because it is sufficiently diverse and complementary to the main discriminative classifier. With one more NFC, i.e., two NFCs, only a small portion of the pseudo-labels can achieve the consensus among these three classifiers. As a result, a large number of samples are discarded even though their pseudo-labels can be true. Too many samples being discarded can cause a performance drop when compared with using a single NFC.

**Sensitivity of the model's performance to hyper-parameter.** The loss weight $\lambda$ for $L_u(\theta_n)$ is the only hyper-parameter in our NorMatch, as in (10). We then evaluate the sensitivity of the classification accuracy (%) to $\lambda$ in Figure 5. Note that the likelihood-based loss $L_u(\theta_n)$ can be much larger than the cross-entropy-based losses in (10), so we tune $\lambda$ from a very small value, e.g., 1e-7. We can see that $\lambda \leq$ 1e-6 can improve the performance of FixMatch + NCUE (the abbreviation of "Normalizing flow for Consensus-based Uncertainty Estimation" proposed in Section 3.2.1), i.e., 92.78% obtained by $\lambda$=0, as is in the 2nd row of Table 1. Note that the performance of $\lambda = 0$ is not drawn in the log-scale plot. While a large $\lambda$ (>1e-6) results in the NUM (i.e., "Normalizing flow for Unsupervised Modeling" proposed in Section 3.2.2) dominating the training process, which may harm the supervised discriminative feature learning and decrease the performance. Hence, we recommend properly setting $\lambda$ so that the $\lambda L_u(\theta_n)$ is smaller than the supervised loss — this constrains $\lambda$ from being very large.

**Evaluation on distribution alignment.** We follow the latest works (Berthelot et al., 2019a; Li et al., 2021) to apply distribution alignment to $\hat{y}_i^u$ (neither sharpened nor one-hot version) in (4). We further evaluate this strategy in Table 5. We observe that the distribution alignment brings 0.3% improvement over the one-hot version (the first two rows). When fully trained for 1024 epochs, our NorMatch with distribution alignment further obtains 94.70%. We thus use it as our final model to compare with the state-of-the-art methods in Section 4.3.

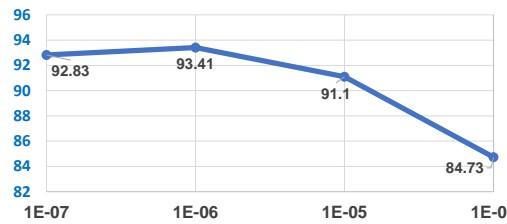

Figure 5: Sensitivity of the model's performance to $\lambda$.

Table 5: Evaluation on distribution alignment (DA).

| Methods | Epochs | Accuracy |
|---|---|---|
| NorMatch w/o DA | 300 | 93.41 |
| NorMatch w/ DA | 300 | 93.71 |
| NorMatch w/ DA | 1024 | **94.70** |

## 4.3 Comparison with the State of the Art

Table 6 reports the comparison of our NorMatch to the other state-of-the-art methods on CIFAR-10, CIFAR-100 and STL-10. We observe that NorMatch achieves the best performance on almost all the label splits,

Table 7: Classification accuracy (%) on Mini-ImageNet.

| Methods | 4000 labels |
|---|---|
| Mean Teacher (Tarvainen & Valpola, 2017) | 27.49 |
| Label Propagation (Iscen et al., 2019) | 29.71 |
| PLCB (Arazo et al., 2020) | 43.51 |
| MixMatch (Berthelot et al., 2019b) | **50.21** |
| SimPLE (Hu et al., 2021) | 49.39 |
| FixMatch (Sohn et al., 2020) | 40.27 |
| NorMatch (Ours) | 48.36 |

favorably outperforming the baseline, FixMatch (Li et al., 2021), and the state-of-the-art methods such as Mean Teacher (Tarvainen & Valpola, 2017), MixMatch (Berthelot et al., 2019b), and CoMatch (Li et al., 2021). Below we analyze the results on each dataset in more detail.

**Results on CIFAR-10 and CIFAR-100.** NorMatch surpasses FlowGMM Izmailov et al. (2020), which uses a single FlowGMM (without the discriminative classifier) to enforce a consistency regularization, by about 15% on CIFAR-10 in the setting of 4000 labels. The better performance demonstrates the effectiveness of introducing the Normalizing Flow Classifier (NFC) to help estimate the uncertainty of pseudo-labels for the discriminative classifier. In addition, our NorMatch is superior to the latest methods, CoMatch (Li et al., 2021) and SemCo (Nassar et al., 2021), e.g., 1.61% over CoMatch on CIFAR-10 in the 40 labels setting and 5.34% over SemCo on 2500 labels of CIFAR-100. The superiority of NorMatch shows that a fundamentally different but complementary NFC for uncertainty estimation is better than an extra discriminative classifier for co-training (SemCo) or an additional projection head for self-training (CoMatch). Compared to Dash (Xu et al., 2021), which employs an adaptive threshold, our NorMatch is threshold-free and with the best performance on the 40 labels of CIFAR-10 and on CIFAR-100 for all label counts. This supports our argument that the threshold-free NorMatch is simpler yet more effective.

NorMatch does not outperform SemCo or Dash on CIFAR-10 in the setting of 250 or 4000 labels. However, we also observe that with these two label counts we reach a near fully supervised performance, hence the performance is saturated as more labels are added. Notably, in a very ideal setting where all images are labelled for training, our fully supervised baseline obtains an accuracy of 95.44%, which is still lower than that of SemCo or Dash. Nevertheless, NorMatch still obtains a performance comparable to SemCo or Dash in these two label counts.

**Results on STL-10.** NorMatch outperforms all the other competitors by at least 1.58%. Notably, it beats the baseline method, FixMatch, by 16%. This significant improvement strongly supports the effectiveness of our NorMatch. Thanks to the NCUE and NUM (see Section 3.2.1 and 3.2.2), NorMatch also excels CoMatch considerably. Moreover, our NorMatch is much simpler than CoMatch because no threshold needs to be tuned and no graph needs to be constructed for contrastive learning.

**Results on Mini-ImageNet.** We further evaluate NorMatch on the challenging Mini-ImageNet, and the results are displayed in Table 7. Our NorMatch achieves significant improvement over the classical Mean Teacher (Tarvainen & Valpola, 2017) and Label Propagation (Iscen et al., 2019) methods. It is also clearly better than the FixMatch baseline (Sohn et al., 2020) by 8.09%, owing to the Normalizing Flow Classifier (NFC) for better uncertainty estimation for pseudo-labels. The better performance illustrates the scalability of our method on the challenge dataset. Furthermore, NorMatch is simpler than the other state-of-the-art methods, with fewer hyper-parameters, especially without the sensitive threshold.

Our NorMatch is on par with PLCB (Arazo et al., 2020) and SimPLE (Hu et al., 2021) but worse than MixMatch (Berthelot et al., 2019b) probably because our baseline method, FixMatch, is much worse than MixMatch, even the large improvement (8.09% over the FixMatch baseline) obtained by NorMatch cannot eliminate the huge gap to MixMatch.

## 5    Limitations and Discussions

Though effective, the NFC in our NorMatch inevitably introduces more computational cost, i.e., 0.08M parameters (see Table 2) and 0.08M Multiply ACcumulate operations (MACs). In addition, the training can fail if we do not stop the gradient of NFC to the backbone CNN, as shown in Table 3. Therefore, we usually need elaborate designs, e.g., gradient stop strategy, to ensure the effectiveness of the auxiliary NFC. Furthermore, the success of our NorMatch largely relies on the baseline, FixMatch, so it can be inferior to other state-of-the-art methods when FixMatch performs poor, e.g., on the challenging Mini-ImageNet dataset as shown in Table 7. On the other hand, as a pseudo-labeling-based method, our NorMatch can hardly boost the performance for a large gap when the FixMatch can already achieve satisfying results, such as 250 and 4000 labels on CIFAR-10 (see Table 6). This is because the noise in pseudo-labels cannot be thoroughly eliminated, which may hinder our NorMatch from achieving a near-saturated classification accuracy.

Despite the above limitations, our NorMatch improves the performance of baseline FixMatch considerably in most circumstances. Notably, when FixMatch achieves comparable results to the other state-of-the-art methods, our NorMatch can outperform the competitors favorably owing to the significant improvement over FixMatch.

## 6    Conclusion

In this paper we propose a novel SSL method called NorMatch. NorMatch leverages a normalizing flow classifier (NFC) to help estimate pseudo-label uncertainty for training a discriminative classifier. This is achieved by applying a Normalizing flow for greeting a Consensus-based Uncertainty Estimation (NCUE) scheme. NCUE evaluates the consensus of the predictions from NFC and the discriminative classifier, then highlights these consistently predicted pseudo-labels and discounts low-confidence ones that cause disagreement. Moreover, NorMatch exploits Normalizing flow for Unsupervised Modeling (NUM), which models the distribution of unlabeled data for better performance. Extensive experiments on CIFAR-10, CIFAR-100, STL-10, and Mini-ImageNet demonstrate that NorMatch achieves state-of-the-art performance.

## Acknowledgements

ZD, AIAR and CBS acknowledge support from the EPSRC grant EP/T003553/1. AIAR acknowledges support from CMIH and CCIMI, University of Cambridge. CBS acknowledges support from the Philip Leverhulme Prize, the Royal Society Wolfson Fellowship, the EPSRC advanced career fellowship EP/V029428/1, EPSRC grants EP/S026045/1 and EP/T003553/1, EP/N014588/1, EP/T017961/1, the Wellcome Innovator Awards 215733/Z/19/Z and 221633/Z/20/Z, the European Union Horizon 2020 research and innovation programme under the Marie Skodowska-Curie grant agreement No. 777826 NoMADS, the Cantab Capital Institute for the Mathematics of Information and the Alan Turing Institute.

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
