# OpenReview forum: "NorMatch: Matching Normalizing Flows with Discriminative Classifiers for Semi-Supervised Learning"
_TMLR — Accepted by TMLR_

### Review · Reviewer_Br9c · 2023-11-23

**Summary Of Contributions:**

This paper proposes a new semi-supervised learning method named NorMatch, which leverages an additional classifier through normalizing flow to estimate the uncertainty of pseudo-labels. NorMatch further employs a diversity-promoting term through normalizing flow as a regularization term. Results on several standard semi-supervised learning datasets validate the effectiveness of NorMatch.

**Audience:**

Yes

**Claims And Evidence:**

Yes

**Requested Changes:**

Generally, this paper meets the standard of TMLR with a new SSL method.

As stated before, the authors need to pay more attention to the writing to improve the readability of this paper. (There are so many concepts in the method section and the results are not well displayed in the experiment section)

**Strengths And Weaknesses:**

Strengths

- the proposed method sounds reasonable

- the results are better than previous SSL methods

- analysis like an ablation study is conducted to verify each component of the proposed method


Weaknesses

- this paper (e.g., the method section and the experiment section) could be further simplified to increase the readability

---

### Review · Reviewer_GUVx · 2023-12-14

**Summary Of Contributions:**

This paper proposes a new uncertainty estimation scheme using normalizing flows, which aims at improving the pseudo-label quality for semi-supervised learning.

**Audience:**

Yes

**Claims And Evidence:**

Yes

**Requested Changes:**

please see the weaknesses

**Strengths And Weaknesses:**

Strengths:
1. The Normalizing Flow Classifier (NFC) could use both the labeled and unlabeled data to learn a better decision boundary.
2. Experiments on four datasets, including the ablation studies and parameter analyses, demonstrate the effectiveness of the proposed method.

Weakness:
1. Is it possible to try another two combinations of the standard classifier and NFC? Namely, what would happen if we use "standard classifier + standard classifier" or "NFC + NFC", compared with the current "standard classifier + NFC" strategy?
2. I encourage the authors to discuss the potential limitations of the proposed NFC to give the readers a more comprehensive understanding of this paradigm.
3. Why the baselines on Mini-ImageNet are less than those on CIFAR and STL? These baselines need to be supplied.
4. How much additional computational cost would the NFC bring?

---

### Review · Reviewer_mhzF · 2023-12-19

**Summary Of Contributions:**

This paper presents a method called NorMatch for semi-supervised learning based on FixMathch method. Normatch introduces a new uncertainty estimation scheme based on the Normalizing Flow Classifier as called in the paper. This scheme acts as an auxiliary classifier to enforce highly certain pseudo-labels, which improves the pseudo quality of discriminative classifiers. The NorMathch also introduces unsupervised learning by maximizing the log-likelihood of unlabeled data. The experiments show that this could further improve the accuracy in SSL tasks. Overall, the main contributions of this paper is by modeling the Gaussian probibilty model in the latent features space to enchance the uncertainty estimation and representation learning.

**Audience:**

Yes

**Claims And Evidence:**

Yes

**Requested Changes:**

- The Normalizing Flow should be introducted and linked to the method with rigorously formal.

- When there are many methods for pseudolabels' purification, the paper should review them more comprehensively and compare them with  analytical perspectives.

**Strengths And Weaknesses:**

## Strengths
- Two techinical parts are presented in this paper: a new psudolabels' purification method when applied to FixMatch and a unsupervised learning loss of NUM. Experiments show that these two skills help to improve the baseline method of FixMatch.
- The presentation is easy to follow with well organization.

## Weaknesses
- But my frist question is that how Normalizing flows act in this manner. Normalizing flow is a generative model by transfering random variable to different representation spaces in order to facilitate the computing of probability mass. In my point, the utilizing the Gaussian probability model in the latent space does not mean it is a Normalizing flow model. The name of Normalizing flow classifier and the NUM are not presented properly in my view. Could you explain why they are so called?

- Second, what are the befinits from other pseudolabels' enhancement methods? since there are many works for doing so, I think the authours should review them more compresensively and study different effects among different strategies of pseudolabels' purification?

---

### Decision · Action_Editor_piZ7 · 2024-01-25

**Recommendation:** Accept as is

**Comment:**

The reviewers found the paper “easy to follow” (Reviewer mhzF), the experiments demonstrate the effectiveness of the proposed method (Reviewer GUVx), the results are better than previous SSL methods and ablations are conducted (Reviewer Br9c). During the rebuttal, the authors have made clarification to the paper to address feedback from the reviewers, added more explanations, baselines and a discussion of limitations. In my opinion, the updated draft meets the standard for TMLR acceptance: it tackles an important problem that is of interest to the community, presents insights and a method that works better than previous SSL methods on some datasets, and as far as I can tell, is technically sound and provides convincing evidence for the claims made.

I encourage the authors to add to the limitations section that some other methods outperform NorMatch on some datasets and perhaps any insights they have on when NorMatch is expected to have an advantage over those methods.

**Audience:**

Yes, this paper is on the topic of semi-supervised learning, presents a new method, results on various datasets and ablations that would interest the community working on this problem area.

**Claims And Evidence:**

This paper proposes an approach to semi-supervised learning. Specifically, they operate in the framework of pseudo labeling the unlabeled data, but introduce a weighting scheme for examples based on the confidence of their pseudolabels, where the confidence is determined based on the agreement of two types of classifiers: a discriminative one and a generative one (based on a GMM obtained via a normalizing flow from the embedding space of a CNN feature extractor).

The authors show that, for some datasets, their approach has better performance than previous baselines and methods and they also conduct ablation studies to understand the effect of different components of their system on that performance. I view this experimental investigation as sufficient to substantiate their claims on improved performance across different datasets ("comparable or better than state-of-the-art"), as well as claims about the necessity and role of the different components of their system. (Note that e.g. on mini-imagenet, after adding more methods suggested by the reviewers, their approach performs slightly worse than SOTA, but I think it's fair to say it's "comparable").

---

> ### Author Response · Authors · 2024-02-13
> **Response to Action Editor piZ7**
>
> Thanks for the suggestions. We have updated "Sec. 5 Limitations and Discussions" to discuss in which situation some other methods can outperform NorMatch and when NorMatch is expected to have an advantage over those methods.